# CAN: A SIMPLE, EFFICIENT AND SCALABLE CONTRASTIVE MASKED AUTOENCODER FRAMEWORK FOR LEARNING VISUAL REPRESENTATIONS

## ABSTRACT

We introduce CAN, a simple, efficient and scalable method for self-supervised learning of visual representations. Our framework is a minimal and conceptually clean synthesis of (C) contrastive learning, (A) masked autoencoders, and (N) the noise prediction approach used in diffusion models. The learning mechanisms are *complementary* to one another: contrastive learning shapes the embedding space across a batch of image samples; masked autoencoders focus on reconstruction of the low-frequency spatial correlations in a single image sample; and noise prediction encourages the reconstruction of the high-frequency components of an image. The combined approach results in a robust, scalable and simple-to-implement algorithm. The training process is symmetric, with $50\%$ of patches in *both views* being masked at random, yielding a considerable efficiency improvement over prior contrastive learning methods. Extensive empirical studies demonstrate that CAN achieves strong downstream performance under both linear and finetuning evaluations on transfer learning and robustness tasks. For instance, when pre-training ViT-B models on the curated ImageNet dataset, CAN achieves $74.8\%$ top-1 linear probing accuracy, an absolute improvement of $6.8\%$ over MAE and $1.3\%$ over SimCLR with the same architecture and data augmentations. CAN is especially useful for pre-training on larger uncurated datasets such as JFT-300M: for linear probe on ImageNet, CAN achieves $75.4\%$ compared to $73.4\%$ for SimCLR and $64.1\%$ for MAE. Finetuning our ViT-L model on ImageNet attains $86.1\%$ top-1, compared to $85.5\%$ for SimCLR, and $85.4\%$ for MAE. The overall FLOPs load of SimCLR is $70\%$ *higher* than CAN for ViT-L models[1].

## 1 INTRODUCTION

Self-supervised learning promises continued advances in the state of the art by enabling the use of increasingly large models and datasets. However, interest in larger datasets has precipitated an increased reliance on web-scraped data collection processes, which result in heterogeneous and "uncurated" datasets (Yu et al., 2022; Radford et al., 2021; Jia et al., 2021). Extreme image heterogeneity has made scaling vision models to uncurated datasets a non-trivial challenge (Tian et al., 2021; Cole et al., 2022). There are two families of self-supervised methods for images which have both proven highly effective on curated datasets (e.g., ImageNet), and are therefore natural candidates for scaling to large, uncurated data. First, masked image models such as the masked autoencoder (MAE) (He et al., 2022) are a nascent set of methods based on a mask-and-reconstruct training mechanism. This classical idea (Ballard, 1987) is enjoying a rejuvenation thanks to favourable efficiency when combined with the vision transformer architecture (Dosovitskiy et al., 2021b). Second, contrastive learning (van den Oord et al., 2018; Chen et al., 2020b; He et al., 2020) trains an encoder to distinguish between pairs of positive samples generated with data augmentations and negative pairs sampled at random. Both approaches have proven to be very powerful self-supervised methods.

Contrastive learning and masked autoencoders (MAE) employ very different learning mechanisms: the former train the encoder to be invariant to semantics-preserving data variations, while MAEs learn spatial statistical correlations. Furthermore, MAE methods treat each sample independently in

---

[1]Code will be released soon.

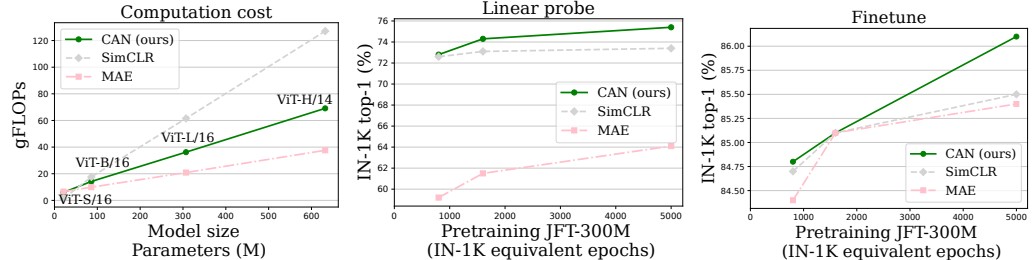

Figure 1: **Left:** CAN scales better than SimCLR since it uses masked inputs. **Middle and right:** CAN outperforms SimCLR and MAE on ImageNet linear probe and finetune evaluations for ViT-L models when pre-training on uncurated data such as JFT-300M.

the loss function, while contrastive methods explicitly look at the relationship between all samples in the batch, by either reducing or increasing embedding distance. Given this, we hypothesize that these two approaches are *complementary*, extracting different discriminative features for a given input. If this hypothesis holds, then we expect to see improved performance on various downstream tasks based on the extracted features. This motivates our exploration of a combined method.

Further, inspired by advances in diffusion models (Ho et al., 2020; Song et al., 2021), we introduce a third loss based on *noise prediction* during the masked autoencoder reconstruction. We add Gaussian noise to unmasked input patches, and train the model to predict the *noise* added to each patch. Denoising encourages the encoder to extract higher-frequency information from the input, while autoencoder reconstructions tend to focus on low-frequency information (Hou et al., 2017). This additional loss has two purposes: it improves downstream performance; and it addresses a source of wasted computation in MAE with a negligible impact on FLOPs: that reconstruction of unmasked patches is thrown away unused.

Combining these ingredients we present CAN, a minimal fusion of contrastive learning, masked autoencoders and denoising diffusion training loss. Our method enjoys stronger performance than its constituent parts do on their own, especially pronounced benefits on more uncurated datasets such as JFT-300M, which contains 300 million highly heterogeneous images, often containing artifacts (e.g., watermarks). For instance, evaluating JFT-trained ViT-L models using the top-1 accuracy of an ImageNet-trained linear probe, MAE achieves 64.1% and SimCLR achieves 73.4%, while CAN achieves 75.4%. CAN masks 50% of patches in each view, making it significantly more scalable than prior contrastive methods that use two full image views. Our contributions are:

1. We present CAN, a simple self-supervised learning algorithm with good scaling properties, making it suitable for training on very large image datasets, such as the JFT-300M dataset.

2. CAN is much more efficient than SimCLR (Figure 1). For instance, SimCLR uses 70% more FLOPs than CAN with ViT-L models.

3. CAN is more robust to distribution shifts than MAE or SimCLR, and performs better on a wide range of few-shot and linear transfer tasks.

## 2 RELATED WORK

**Masked image models with Vision Transformers.** The advent of the Vision Transformer (ViT) (Dosovitskiy et al., 2021b) provoked a focused effort to develop strong self-supervised learning frameworks that use ViT backbones. Works such as DINO (Caron et al., 2021) and MoCo-v3 (Chen et al., 2021b) demonstrated that techniques developed with ConvNet backbones in mind could also perform competitively using ViTs after proper tuning to suit the new architecture. ViT-specific methods have emerged since then, particularly masked image modelling (Bao et al., 2022; Chen et al., 2022; Xie et al., 2022), which takes inspiration from pre-training methods used in NLP (Devlin et al., 2018). Notably MAE (He et al., 2022) showed that classical masked autoencoding approaches could be used to pre-train ViTs *without* passing masked tokens through the encoder. This provides a significant efficiency boost; our method similarly takes advantage of this.

**Contrastive learning in computer vision.** Self-supervision has received significant attention in computer vision as it offers a way to extract general purpose features without supervision. In particular, contrastive learning (van den Oord et al., 2018; Hénaff et al., 2020; Chen et al., 2020b; He et al., 2020; Tian et al., 2020; Chuang et al., 2020; Hénaff et al., 2021) has achieved state of the art performance by enforcing invariance to augmentations, whilst using negative samples (Robinson et al., 2021a; Ge et al., 2021) to avoid trivial solutions by spreading the embedding out uniformly on the sphere (Wang & Isola, 2020). The contrastive pre-training task is conceptually very different from masked image models such as MAE, which learn spatial statistical dependencies. Another distinction is that autoencoders encourage information preservation in latent representations, whilst contrastive learning could suppress features (Chen et al., 2021a; Robinson et al., 2021b). This leads us to hypothesize that the two approaches learn different data features, and may therefore be complementary learning mechanisms. This motivates us to combine contrastive learning and masked image modelling so as to develop a reinforced pre-training task that enjoys the merits of each.

**Denoising diffusion models.** Denoising autoencoders (DAE) (Vincent et al., 2010) learn to reconstruct clean data given a noisy input. By learning to map low-density data regions to high-density regions, DAE learns the shape of the data manifold. This connection was made precise by Vincent (2011), who showed that DAEs learn the score-function $s(\mathbf{x}) = \nabla_{\mathbf{x}} \log p(\mathbf{x})$. This key observation underpins the significant recent advances in generative diffusion models, which use an estimate of the score-function to generate samples (Ho et al., 2020; Song et al., 2021). The recent success of DAEs in generative modelling has not yet translated to representation learning, with some exceptions (Asiedu et al., 2022; Zaidi et al., 2022). In this work we exploit a denoising autoencoder to eliminate the MAE inefficiency of reconstructing unmasked patches but never using them.

**Concurrent work.** Several recent works propose approaches that combine ideas from masked image modelling and Siamese self-supervised learning. For instance, Huang et al. (2022) propose a combination of contrastive and masked reconstruction objectives using one masked view, and one full (unmasked) view. Other recent works (Tao et al., 2022; Chen et al., 2022; Assran et al., 2022) use similar asymmetric designs. The key distinction between CAN and concurrent work is that we strike a different balance between simplicity, efficiency, and performance: we focus on developing a *simple*, *efficient* and symmetric method: we use *two masked views* and no momentum encoder. We hope the simplicity and efficiency of CAN will make it easy to adapt and modify in future work.

## 3 A SIMPLE CONTRASTIVE MASKED AUTOENCODER FRAMEWORK

Our approach is a minimal synthesis of contrastive learning, the masked autoencoder (He et al., 2022), and the denoising loss used in the training of diffusion models. We focus on simplicity and scalability, aiming to design a hybrid with as few complex or costly components as possible. We also aim to minimize *wasted* computation: in particular, the MAE decoder requires reconstructions of all patches, but only those of masked patches are used in the loss. Below, first we detail the basic pipeline of generating views and passing masked inputs through the encoder and decoder. Then we explain the three different objectives we use: contrastive, reconstruction, and denoising. The penultimate section describes the combined objective, and the final section discusses scalability.

### 3.1 OVERVIEW OF METHOD

Given a batch of $n$ images $\{\mathbf{x}\}_{i=1}^n$, we generate two views $\mathbf{x}_i^1, \mathbf{x}_i^2 \in \mathbb{R}^{h \times w \times 3}$ of each image without supervision using the same data augmentations as Chen et al. (2020b). Each image is then split into $T = (h/p) \times (w/p)$ patches of size $p \times p$: $\mathbf{x}_{i,\text{patch}}^1, \mathbf{x}_{i,\text{patch}}^2 \in \mathbb{R}^{T \times p \times p \times 3}$ in preparation for input to the ViT encoder. We always assume that $p$ divides $h$ and $w$. Two masks $\mathbf{M}_i^1, \mathbf{M}_i^2 \in \{0,1\}^T$ are independently generated, with a 1 in coordinate $t \in \{1, \ldots T\}$ indicating that the $t$th patch is masked. Each patch is left unmasked independently with probability $m$, conditioned on always having exactly $T' = m \cdot T$ patches unmasked, which we assume is an integer. In all CAN experiments our default masking rate is $m = 50\%$ unless explicitly stated otherwise (for MAE we use the default 75%). Following He et al. (2022), only the $T'$ *unmasked* patches are passed to the ViT encoder, which processes the two views in parallel. Masking a large fraction of patches from both views make our method much more efficient (see Table 1) than contrastive methods that use two full views, and recent works that use one full view and one masked view (Assran et al., 2022; Huang et al., 2022). Finally, we collect the embeddings of unmasked tokens $\mathbf{z}_i^1, \mathbf{z}_i^2 \in \mathbb{R}^{T' \times d}$ and

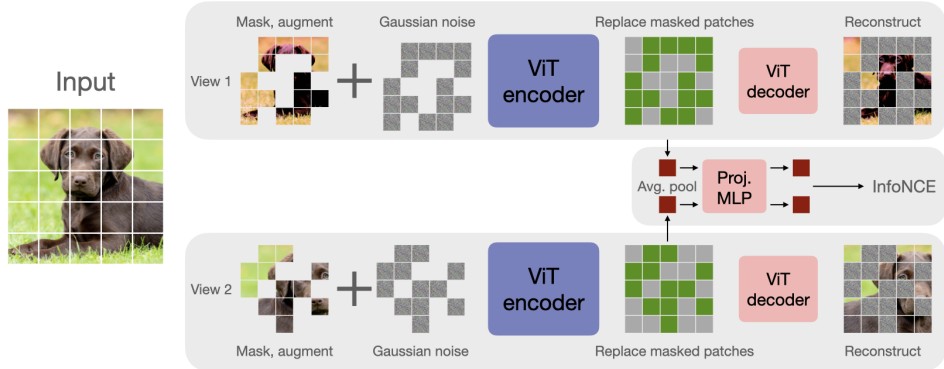

Figure 2: **The CAN framework:** Two views of an image are generated, $50\%$ of patches randomly masked in each, and noise is added to patches. An encoder is trained to solve three tasks: 1) **Reconstruction:** encoded patches are passed to a decoder that reconstructs missing patches, 2) **Denoise:** reconstructs the noise added to unmasked patches, and 3) **Contrast:** pooled patches are passed to a contrastive loss, using in-batch samples as negatives (Chen et al., 2020b).

reshape into $T \times d$ tensors by adding a learned [M] embedding to positions corresponding to masked tokens. The result is passed through a comparatively lightweight ViT decoder to produce outputs $\hat{\mathbf{x}}_i^1, \hat{\mathbf{x}}_i^2$ in image space $\mathbb{R}^{h \times w \times 3}$.

## 3.2 CONTRASTIVE LEARNING OBJECTIVE

The embeddings $\mathbf{z}_i^1, \mathbf{z}_i^2 \in \mathbb{R}^{T' \times d}$ returned by the encoder are pooled via a simple mean along the first dimension to form $d$-dimensional embeddings, which are passed through a lightweight MLP projection head that maps into a lower dimension space $\mathbb{R}^r$, $r < d$, and normalized to unit length to produce embeddings $\mathbf{u}_i^1, \mathbf{u}_i^2 \in \mathbb{R}^r$ for $i = 1, \ldots n$. For the $i$th batch item we collect the other $2n - 2$ samples in-batch $\mathcal{N}_i = \{\mathbf{u}_j^1, \mathbf{u}_j^2\}_{j \neq i}$ to use as negatives, and compute the InfoNCE loss:

$$\mathcal{L}_{\text{InfoNCE}} = \frac{1}{2n} \sum_{v=1,2} \sum_{i=1}^{n} -\log \frac{e^{\mathbf{u}_i^{1\top} \mathbf{u}_i^2 / \tau}}{e^{\mathbf{u}_i^{1\top} \mathbf{u}_i^2 / \tau} + \sum_{\mathbf{u}^- \in \mathcal{N}_i} e^{\mathbf{u}_i^{v\top} \mathbf{u}^- / \tau}}$$

where $\tau > 0$ is a temperature parameter, which we set to $\tau = 0.1$ by default.

## 3.3 PATCH RECONSTRUCTION OBJECTIVE

The outputs $\hat{\mathbf{x}}_i^1, \hat{\mathbf{x}}_i^2$, $i = 1, \ldots, n$ of the ViT decoder are trained to reconstruct the missing patches of each image. Corroborating the findings of He et al. (2022), we find it best to only compute the reconstruction loss on masked patches:

$$\mathcal{L}_{\text{rec}} = \frac{1}{2n} \sum_{v=1,2} \sum_{i=1}^{n} \|\mathbf{M}_i^v \circ (\mathbf{x}_i^v - \hat{\mathbf{x}}_i^v)\|_2^2$$

where $\circ$ multiplies all pixels in the $t$th patch of the residual image $\mathbf{x}_i^v - \hat{\mathbf{x}}_i^v$ by $(\mathbf{M}_i^v)_t \in \{0, 1\}$. Whilst computing the loss only on masked patches gives better performance, it indicates wasted computation since the decoder also produces reconstructions for unmasked patches. To avoid waste we propose an alternative objective specifically for unmasked patches, which we discuss next.

## 3.4 DENOISING OBJECTIVE

Inspired by the significant advances in diffusion modelling using *denoising* training objectives (Ho et al., 2020; Kingma et al., 2021) and their equivalent score-based counterparts (Song et al., 2021; Vincent, 2011) we revisit the suitability of denoising for self-supervised learning. We add independent isotropic Gaussian noise to each image $\mathbf{x}_i^v \leftarrow \mathbf{x}_i^v + \sigma_i^v \mathbf{e}_i^v$ with $\mathbf{e}_i^v \sim \mathcal{N}(\mathbf{0}, I)$ and $\sigma_i^v$ uniformly

sampled from an interval $[0, \sigma_{\max}]$. This noisy input is masked and passed to the encoder as described in Section 3.1. When passing encoded patches to the decoder we make a small addition to the method in Section 3.1 to provide the decoder with information on the noise level $\sigma_i^v$ to help it separate noise from the ground truth image. This is motivated by denoising diffusion methods, which pass both the noisy image and the noise level as inputs to the denoising model (Ho et al., 2020). We achieve this by using $\sigma_i^v$ as a positional encoding in the decoder, similarly to Vaswani et al. (2017). First we produce a sinusoidal embedding of $\sigma_i^v \in \mathbb{R}^d$, which is passed through a lightweight 2 layer MLP with ReLU activations of constant width $d$ to produce a (learnable) embedding $\mathbf{p}_i^v \in \mathbb{R}^d$, whose dimension matches the latent dimension of $\mathbf{z}_i^v \in \mathbb{R}^{T \times d}$. We add the result to each embedded token (including missing tokens [M]) to provide noise-level information: $(\mathbf{z}_i^v)_t \leftarrow (\mathbf{z}_i^v)_t + \mathbf{p}_i^v$ for $t = 1 \ldots, T$, and pass the result to the decoder producing $\mathbf{x}_i^v$. We define our denoising loss function, which is computed only on unmasked pixels:

$$\mathcal{L}_{\text{denoise}} = \frac{1}{2n} \sum_{v=1,2} \sum_{i=1}^{n} \|(1 - \mathbf{M}_i^v) \circ (\sigma_i^v \mathbf{e}_i^v - \hat{\mathbf{x}}_i^v)\|_2^2$$

where, $\circ$ is as defined in Section 3.2. Note that this denoising loss is extremely lightweight, introducing only a very small overhead due to the MLP. We emphasize that the reconstruction of noise patches comes at zero additional cost since the decoder produces reconstructions of all patches, both masked and unmasked, even though only reconstructions of masked patches are used in $\mathcal{L}_{\text{rec}}$. Finally, it has often been observed in the diffusion modelling literature that

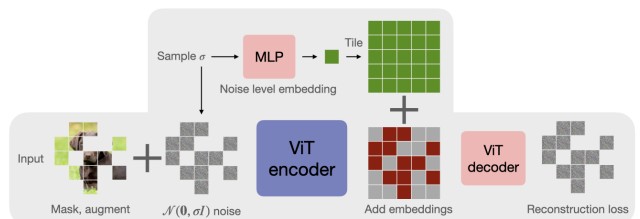

Figure 3: **Denoising:** Both the encoded patches and the noise level $\sigma$ are passed to the decoder by passing $\sigma$ through an MLP, and adding the result to each embedded token.

although it is equivalent to train a denoising model to estimate the noise, or to estimate the clean input itself (Vincent, 2011), there is a big empirical gap between the two, with noise prediction faring better. While we do not pursue it further, our testing corroborates this.

**Ablation:** Table 1 studies the effect of each of the components of the denoising method. We use ViT-B models trained for 100 epochs on ImageNet, and consider four settings, each adding in more parts of the method: 1) CAN with no denoising, 2) adding noise to the input only, 3) adding noise and using the denoising loss, and 4) the full method with all of the described

| None | +noise | +noise, +loss | Full |
|---|---|---|---|
| 67.9 | 68.6 | 68.4 | 68.9 |

Table 1: Ablating components of the denoising objective. "Full" denotes the entire method as described in Section 3.4

components, including using $\sigma_i^v$ as a positional encoding in the decoder. Results show that simply adding noise as a data augmentation improves performance by 0.7%, which can be improved to 1% by adding a reconstruction loss with noise level passed as an argument. The noise level argument is necessar: the reconstruction loss without noise level argument performs worse (68.4%) than noise with no reconstruction at all (68.6%).

We emphasize that the improvement from denoising comes at minimal cost to run time and memory during training, since it uses reconstructions produced by the decoder, which in the case of MAE are simply thrown away unused. Denoising prediction encourages the encoder to extract high-frequency features, which we hypothesize is complementary to reconstruction and contrastive tasks.

### 3.5 THE COMBINED OBJECTIVE FUNCTION

The overall CAN objective trains the encoder and decoder to optimize three losses combined:

$$\mathcal{L}_{\text{CAN}} = \lambda_{\text{InfoNCE}} \mathcal{L}_{\text{InfoNCE}} + \lambda_{\text{rec}} \mathcal{L}_{\text{rec}} + \lambda_{\text{denoise}} \mathcal{L}_{\text{denoise}}$$

where $0 \leq \lambda_{\text{InfoNCE}}, \lambda_{\text{rec}}, \lambda_{\text{denoise}}$, and $\lambda_{\text{InfoNCE}} + \lambda_{\text{rec}} + \lambda_{\text{denoise}} = 1$ weight the objectives. In practice we parameterize the weights by eliminating one variable using the equality constraint, taking: $\lambda_{\text{rec}} = (1 - \lambda_{\text{InfoNCE}}) \cdot \lambda$ and $\lambda_{\text{denoise}} = (1 - \lambda_{\text{InfoNCE}}) \cdot (1 - \lambda)$ where $0 \leq \lambda \leq 1$. This parameterization

makes it easy to control the relative weighting between the two reconstruction losses $\mathcal{L}_{\text{rec}}, \mathcal{L}_{\text{denoise}}$ on the one hand, and the contrastive loss $\mathcal{L}_{\text{InfoNCE}}$ on the other. Empirically we find that performance is very robust to the choice of $\lambda$, and many choices of $\lambda_{\text{InfoNCE}}$ also work well (see Section 5).

### 3.6 DISCUSSION ON EFFICIENCY

The goal of this work is to propose a conceptually minimal combined contrastive masked autoencoder approach, aiming to find better trade-offs between simplicity, efficiency, and performance. Consequently, we choose to omit a number of commonly used self-supervised learning design components. For instance, we do not use a momentum target network or multiple views (multi-crop), since they both increase memory requirements and run time. Even without these commonly used components, our minimal framework achieves very strong performance compared to prior work, and importantly improves performance over its contrastive and autoencoder constituent parts. We expect that a wide range of modifications, such as momentum target networks (He et al., 2020) and multi-crop (Caron et al., 2020), will improve performance further on top of the core method.

## 4 RESULTS

### 4.1 PRE-TRAINING ON UNCURATED DATA: JFT-300M

A key promise of self-supervised learning is to allow models to be trained on extremely large scale image datasets collected from the Web. Not only is such data likely to be *unannotated*, but also *uncurated*: images containing many objects, variable lighting, artifacts (e.g., watermarks) and so on. The large variation in images found online presents a major challenge to self-supervised learning, and it is not guaranteed that methods that work well on curated (and comparatively smaller) datasets such as ImageNet will work equally well on less curated data. To study how CAN scales to large datasets we use JFT-300M (Sun et al., 2017), a dataset of around 300 million images.

| | Architecture | Epochs | IN-1K top-1 |
|---|---|---|---|
| MoCLR (Tian et al., 2021) | R50 | 5000 | 67.6 |
| BYOL (Grill et al., 2020) | R50 | 5000 | 67.9 |
| DnC (Tian et al., 2021) | R50 | 1000 | 67.9 |
| DnC (Tian et al., 2021) | R50 | 4500 | 70.7 |
| MoCLR (Tian et al., 2021) | R200×2 | 5000 | 74.2 |
| DnC (Tian et al., 2021) | R200×2 | 3000 | **77.3** |
| MAE† (He et al., 2022) | ViT-L | 1600 | 50.5 |
| MAE† (He et al., 2022) | ViT-L | 5000 | 64.1 |
| SimCLR† (Chen et al., 2020b) | ViT-B | 800 | 65.8 |
| SimCLR† (Chen et al., 2020b) | ViT-L | 800 | 72.6 |
| SimCLR† (Chen et al., 2020b) | ViT-L | 1600 | 73.1 |
| SimCLR† (Chen et al., 2020b) | ViT-L | 5000 | 73.4 |
| **CAN (ours)** | ViT-B | 800 | 67.1 |
| **CAN (ours)** | ViT-L | 800 | 72.8 |
| **CAN (ours)** | ViT-L | 1600 | 74.3 |
| **CAN (ours)** | ViT-L | 3000 | 75.3 |
| **CAN (ours)** | ViT-L | 5000 | 75.4 |

Table 2: **JFT-300M pre-training:** Comparison to the state of the art on ImageNet linear probe. CAN outperforms all methods except DnC, which uses a complicated multi-stage training process. Computation is measured as ImageNet-equivalent epochs. †Our implementation.

**Setup.** Training time is measured in ImageNet-equivalent epochs: 1 epoch equals $1281167/[\text{batch size}]$ steps, the number of steps in one IN-1K epoch. Models are evaluated using linear probe and finetuning on IN-1K. All hyperparameers were tuned on IN-1K, besides learning rate and weight decay which we cut by a factor of 4 and 2 respectively to stabilize training on JFT-300M. See Appendix C and Section 5 for details.

**Results.** Figure 1 compares CAN to SimCLR and MAE baselines using ViT-L models. CAN achieves a much better trade-off between efficiency (measured in FLOPs) and performance using ViT-L models for all three methods: CAN uses $41\%$ fewer FLOPs than SimCLR and consistently outperforms SimCLR and MAE: for training ViT-L models for 5000 epochs, CAN achieves an IN-1K linear probe performance of $75.4\%$, compared to $71.8\%$ for SimCLR and $64.1\%$ for MAE. The relatively poorer linear probe performance of MAE on JFT-300M highlights the non-triviality of scaling from IN-1K to larger datasets and suggests that while MAE is scalable for *model size*, scalability to larger *datasets* requires further study. Figure 1 (right) gives finetuning results. CAN performs favourably: for a 5000 epoch pre-training schedule, CAN achieves an IN-1K linear probe performance of $86.1\%$, compared to $85.5\%$ for SimCLR and $85.4\%$ for MAE. CAN also enjoys better scaling with training schedule length than either MAE or SimCLR.

| Method | Pre-training epochs | Encoder | No Additional params. | Masked image | Finetune | Linear probe |
|---|---|---|---|---|---|---|
| *from scratch*[†] | 100 | ViT-B | ✓ | ✗ | 79.1 | — |
| *from scratch* | 300 | Swin-T | ✓ | ✗ | 81.3 | — |
| MoCo-v3 (Chen et al., 2021b) | 300 | ViT-B | ✗ | ✗ | 83.0 | 76.7 |
| DINO (Caron et al., 2021) | 1600 | ViT-B | ✗ | ✗ | 82.8 | 78.2 |
| EsViT (Li et al., 2021) | 300 | Swin-T | ✗ | ✗ | — | 78.1 |
| CIM (Fang et al., 2022) | 300 | ViT-B | ✗ | ✗ | 83.1 | — |
| CAE (Chen et al., 2022) | 800 | ViT-B | ✗ | ✗ | 83.8 | 68.6 |
| CAE (Chen et al., 2022) | 1600 | ViT-B | ✗ | ✗ | 83.9 | 70.4 |
| BEiT (Bao et al., 2022) | 800 | ViT-B | ✗ | ✗ | 83.2 | 37.6* |
| SimMIM (Xie et al., 2022) | 800 | ViT-B | ✓ | ✗ | 83.8 | 56.7 |
| MAE (He et al., 2022) | 800 | ViT-B | ✓ | ✓ | 83.1 | — |
| MAE (He et al., 2022) | 1600 | ViT-B | ✓ | ✓ | 83.6 | 68.0 |
| **CAN (ours)** | 800 | ViT-B | ✓ | ✓ | 83.4 | 74.0 |
| **CAN (ours)** | 1600 | ViT-B | ✓ | ✓ | 83.6 | 74.8 |
| SimCLR[†] (Chen et al., 2020b) | 800 | ViT-L | ✓ | ✗ | 83.4 | 73.9 |
| MAE (He et al., 2022) | 800 | ViT-L | ✓ | ✓ | 84.9 | 73.5 |
| MAE[†] (He et al., 2022) | 800 | ViT-L | ✓ | ✓ | 83.7 | 71.4 |
| **CAN (ours)** | 800 | ViT-L | ✓ | ✓ | 84.7 | 76.2 |

Table 3: **Pre-training on ImageNet-1K.** [†]Our implementation. *Quoted from Chen et al. (2022).

We also compare CAN to the current state of the art on JFT-300M pre-training in Table 2. Our best performance, 75.4% with ViT-L outperforms all methods besides DnC, with 77.3% (Tian et al., 2021) with R200×2. However we note that CAN is *considerably* simpler than DnC, which involves multiple training steps including training 10 separate "expert" models (each as large as the final model), and then using MoCLR (an improvement of SimCLR that adds a momentum encoder and more), and finally using distillation to produce a single model. Our calculations suggest that training a ViT-L with CAN is about 3× faster than training the considerably smaller ResNet50 with DnC in terms of wall clock time (see Appendix B for explanation). CAN on ViT-L outperforms MoCLR with R200×2 backbone (similar parameter counts), where we note that MoCLR performs as well or better than BYOL and MoCo-v3 on IN-1K (Tian et al., 2021).

## 4.2 PRE-TRAINING ON IMAGENET

Next we evaluate our method using ImageNet (IN-1K) pre-training to verify that it remains competitive in this setting. Results in Table 3 record the top-1 accuracy on IN-1K classification of finetuned models, and linear probes. Finetuning CAN achieves 83.6% with ViT-B, outperforming other contrastive approaches such as MoCo-v3 (83.0%), and is competitive with other state-of-the-art approaches such as CAE (83.9%). The linear probe performance of CAN is 74.8% using ViT-B, beating all masked image modelling methods, the best of which is CAE with 70.4% (Chen et al., 2022). CAN is only outperformed by MoCo-v3 and DINO, both of which use momentum encoders and two full image views, and in the case of DINO a further 10 multi-crop views. Note that the *masked image* column indicates whether a method uses one or more full image views as input to the model, and the *no additional parameters* column indicates whether a method relies on other parameters besides the main encoder, e.g., from a pre-trained tokenizer, or a momentum updated target encoder. We also report results for our MAE implementation, which approximately matches the original numbers reported by He et al. (2022), validating our MAE results on JFT-300M.

## 4.3 FEW-SHOT LEARNING

We use linear probes to evaluate suitability of CAN for few-shot learning, following the protocol of Dosovitskiy et al. (2021a). We use the models pre-trained on JFT-300M for 5000 epochs whose ImageNet performance is recorded in Figure 1. Results in Figure 4 for few-shot transfer learning on 9 other datasets show that the superior performance on IN-1K translates to strong performance on other tasks. We also note that our 25-shot ViT-L models beat *full-shot* both DnC and BYOL ResNet50 models (also trained for 5000 epochs on JFT-300M) on 6 out of 8 datasets (Tian et al., 2021). See Appendix A for many additional results for different training schedules and model sizes.

## 4.4 ROBUSTNESS TO DISTRIBUTION SHIFT

Finally, we consider the robustness of CAN to distribution shifts. We use ViT-L backbones trained for 5000 epochs on JFT-300M, which have been finetuned on IN-1K. Model performance is evaluated on a number of different validation sets with the same 1000 classes as IN-1K Mao et al. (2022).

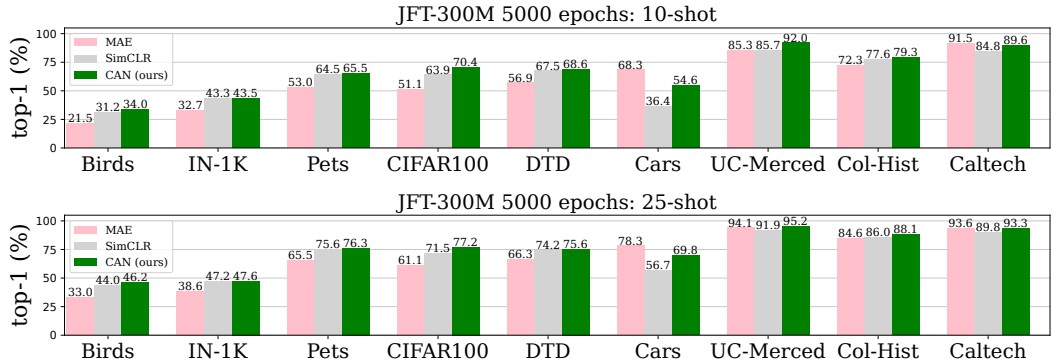

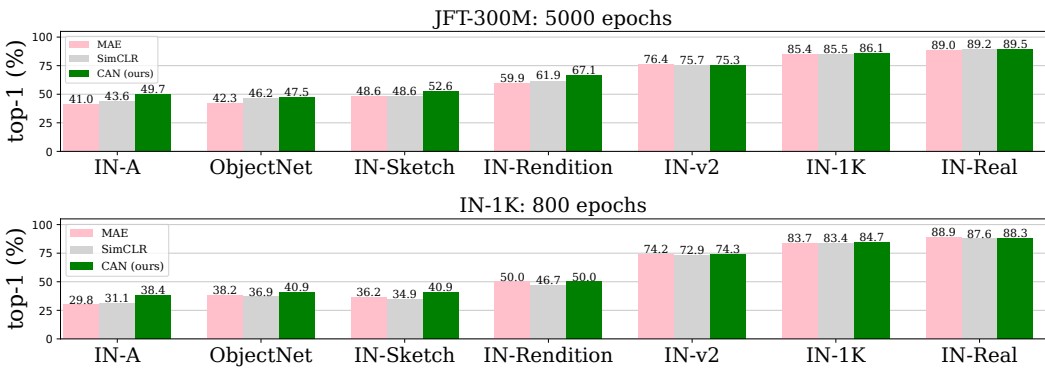

Figure 4: **Few-shot:** ViT-L models pre-trained on JFT-300M for 5000 epochs are evaluated on 9 datasets in few-shot setting (10-shot and 25-shot). CAN outperforms MAE and SimCLR.

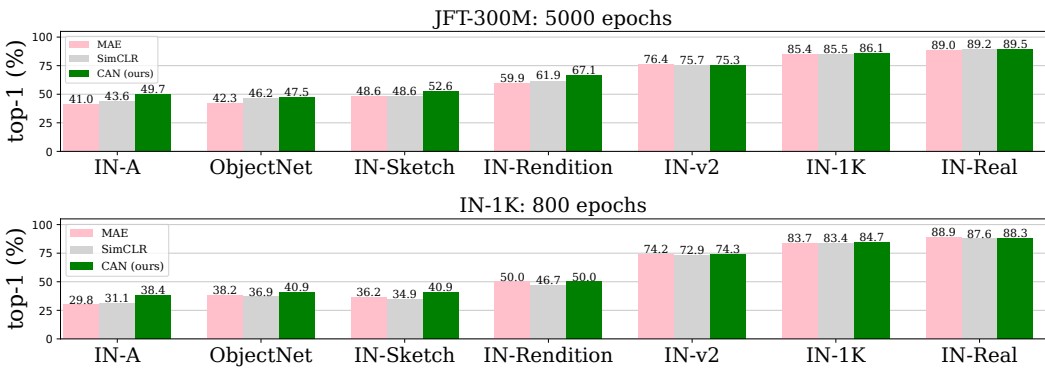

Figure 5: **Robustness:** Evaluating performance under distribution shifts with respect to models finetuned on IN-1K. Validation performance of ViT-L models is reported on 7 different datasets.

Figure 5 reports results on the following 7 validation sets, which cover a large variety of distribution shifts: original IN-1K (Deng et al., 2009), IN-v2 (Recht et al., 2019), IN-ReaL (Beyer et al., 2020), IN-Adversarial (Hendrycks et al., 2021b), IN-Rendition (Hendrycks et al., 2021a), Object-Net (Barbu et al., 2019). CAN performs favourably under both JFT-300M and IN-1K pre-training, beating SimCLR and MAE baselines in nearly all cases. See Appendix A for additional results.

## 5 HYPERPARAMETER ANALYSIS

We study the different components of CAN to better understand the effect of the different mechanisms, and to determine optimal parameter configurations. All ablations use ViT-B models trained for 100 epochs on IN-1K, unless explicitly said otherwise. We use the best loss weights and noise level in these experiments for experiments in Section 4.

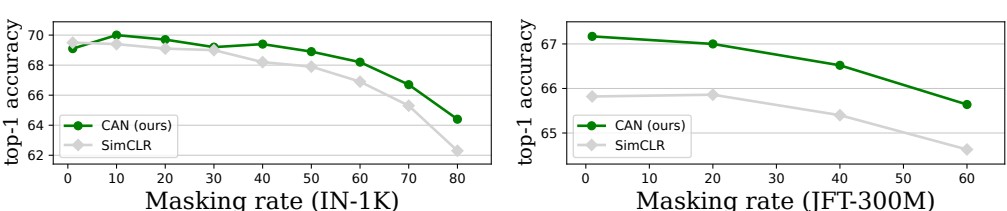

Figure 6: CAN and SimCLR with different masking rates. ViT-B models are pre-trained for 100 epochs on IN-1K (left), and 800 epochs on JFT-300M (right).

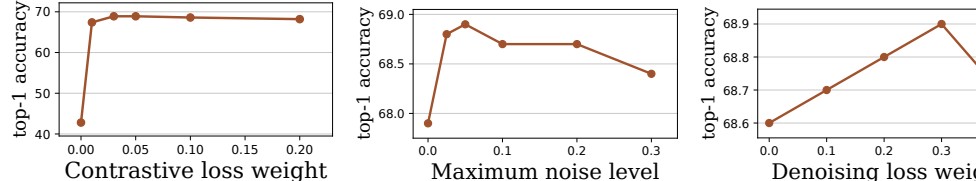

Figure 7: ViT-B models pre-trained on IN-1K for 100 epochs. **Left:** The best contrastive loss weight is small but non-negative. **Middle:** A wide range of $\sigma_{\max}$ values improve over no-noise. **Right:** Performance is not sensitive to the denoising loss weight.

**Complementarity of contrastive and reconstruction losses.** A key hypothesis motivating our work is that contrastive learning and masked autoencoder reconstruction may not only be compatible training objectives, but are *complementary* ones. Table 4 compares the final training value of the contrastive $\mathcal{L}_{\mathrm{InfoNCE}}$ and reconstruction $\mathcal{L}_{\mathrm{rec}}$ when jointly trained (i.e., CAN) compared to only optimizing $\mathcal{L}_{\mathrm{InfoNCE}}$ (SimCLR) or only $\mathcal{L}_{\mathrm{rec}}$ (MAE). The results support the hypothesis: joint training achieves a lower loss on *both* objectives compared to individual training.

**Masking rate.** Figure 6 reports the behavior of CAN and SimCLR under different masking rates on IN-1K and JFT-300M pre-training (for JFT-300M we use 800 epochs). The performance of SimCLR decreases as the masking rate increases, suggesting that masking is not an effective data augmentation. In contrast, performance of CAN peaks at a non-zero masking rate, but at a much lower rate than the $75\%$ used by MAE on IN-1K. This occurs since very low masking rates are preferred by the contrastive part of CAN, but severely damage the autoencoder part, which can learn trivial solutions. The considerable efficiency improvement from masking $50\%$ of patches more than compensates for the small drop in performance for a fixed number of epochs.

| Method | Contrastive loss ↓ | Reconstruction loss ↓ |
|---|---|---|
| SimCLR | 9.157 | — |
| MAE | — | 0.1658 |
| **CAN (ours)** | **9.143** | **0.1633** |

Table 4: **Complementary training:** All methods use $50\%$ masking for fair comparison. CAN training achieves *lower* training loss for both contrastive and reconstruction than individual training.

**Contrastive loss weight.** We vary the weighting $\lambda_{\mathrm{InfoNCE}}$ used to weight the contribution of the contrastive and reconstruction losses. Recall that larger $\lambda_{\mathrm{InfoNCE}}$ places higher weight on the contrastive loss. Results in Figure 7 show that the best weight is $\lambda_{\mathrm{InfoNCE}} = 0.03$, which approximately balances the magnitudes of the two terms (see Table 4).

| Method | ImageNet top-1 ↑ |
|---|---|
| AN | 42.8 |
| CN | 68.5 |
| CA | 67.9 |
| **CAN (ours)** | **68.9** |

**Denoising loss weight and noise level.** We study the noise level interval $[0, \sigma_{\max}]$ from which to sample input noise, and the weight $\lambda$ balancing the denoising and reconstruction losses. Results in Fig 7 show that the best maximum noise level is $\sigma_{\max} = 0.05$, and that similar performance is attained for a number of different weights on the denoising loss.

Table 5: **Ablating CAN:** We remove each of the three loss terms in CAN one by one.

**Ablating CAN**: CAN is comprised of three components: (C) contrastive, (A) masked autoencoder, and (N) de-noising losses. We ablate each of the three components in Table 5, setting the loss weight to zero to "remove" a component. We use ViT-B models pre-trained for 100 epochs. Removing any component leads to worse performance, with contrastive loss hurting the most.

## 6    DISCUSSION

We present CAN, a simple, efficient and scalable self-supervised method for visual representation learning. CAN combines ideas from contrastive learning, masked autoencoding, and diffusion denoising into a single high-performing method. Extensive empirical results show that CAN scales with minimal changes to the large uncurated datasets, providing a significant boost over SimCLR and MAE methods on a wide range of downstream tasks and evaluations, including linear probes, few-shot, robustness, and finetuning. Our results suggests that contrasting and reconstruction are complementary principles that can mutually reinforce one another.

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

## A  ADDITIONAL TRANSFER LEARNING RESULTS

We report additional results for few-shot learning and robustness.

**Robustness:** Sections 4.4 reports robustness results for ViT-L models pre-trained on JFT-300M for 5000 epochs, and ViT-L models pre-trained on IN-1K for 800 epochs. In both cases we report the performance of the models after finetuning on IN-1K.

Here we report the same robustness results for ViT-L models trained on JFT-300M for 1600 and 800 epochs (Figure 8), and ViT-B models pre-trained for 800 epochs (Figure 9). Figure 9 also compares our ViT-B model to ViT-B models trained from scratch on ImageNet. We find that our model is considerably more robust than training with cross-entropy and Mixup from scratch, and also outperforms PyramidAT (Herrmann et al., 2022), an adversarial training method that introduces significant overheads compared to standard cross-entropy training. We emphasize that here there are two differences in the training: a) the training algorithm itself, and b) the data seen by the model. Our model sees extra JFT-300M data not seen by the other two approaches. This means that the methods are not exactly comparable. It is, however, a realistic setting showing the benefits to robustness of pre-training on large datasets.

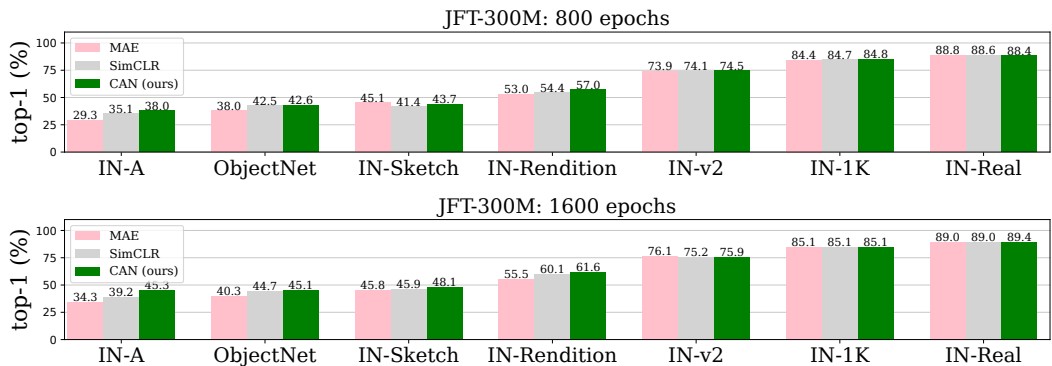

Figure 8: ViT-L models pre-trained on JFT-300M for 800 and 1600 epochs respectively, evaluated on 7 datasets with distribution shifts from IN-1K.

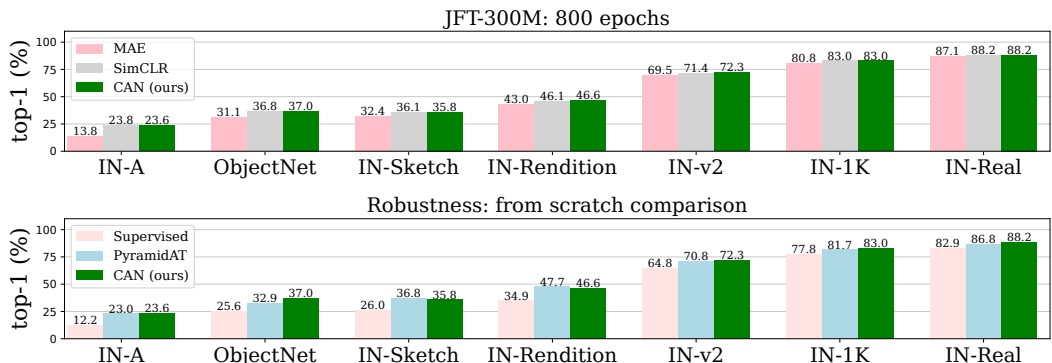

Figure 9: **Top:** ViT-B models pre-trained on JFT-300M for 800 epochs, evaluated on 7 datasets with distribution shifts from IN-1K. **Bottom:** Comparison of our JFT-300M pre-trained ViT-B model to training ViT-B from scratch on IN-1K. We compare to standard supervised cross-entropy training with Mixup, and to PyramidAT (Herrmann et al., 2022), which uses an adversarial training method. CAN considerably outperforms supervised training, and beats PyramidAT in 6 out of 7 cases without requiring adversarial training.

**Few shot:** Section 4.3 reports 10- and 25-shot results for ViT-L models pre-trained on JFT-300M for 5000 epochs. Here we report 1- and 5-shot results for the same models in Figure 10. We additionally

show the full set of $\{1, 5, 10, 25\}$-shot results for ViT-L models pre-trained on JFT-300M for 800 and 1600 epochs (Figures 11 and 12 respectively), ViT-B models pre-trained on JFT-300M for 800 epochs (Figure 13), and ViT-L models pre-trained on IN-1K for 800 epochs (Figure 14).

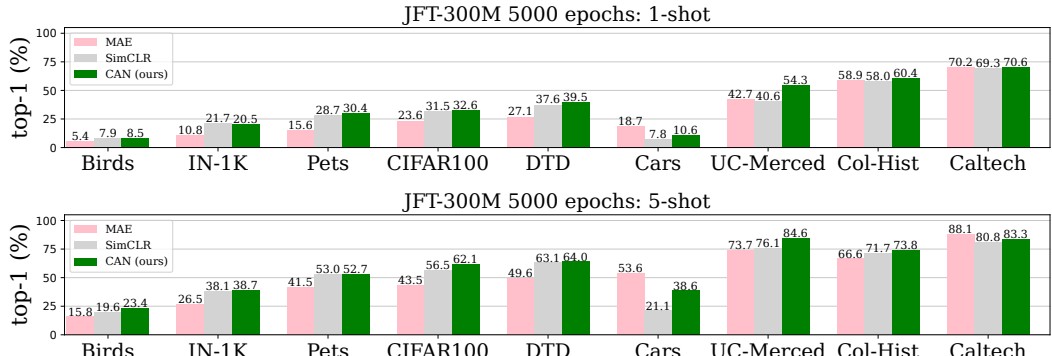

Figure 10: **Few shot:** ViT-L models pre-trained on JFT-300M for 5000 epochs evaluated on 9 few-shot learning tasks. Results accompany the 10- and 25-shot results in Figure 4.

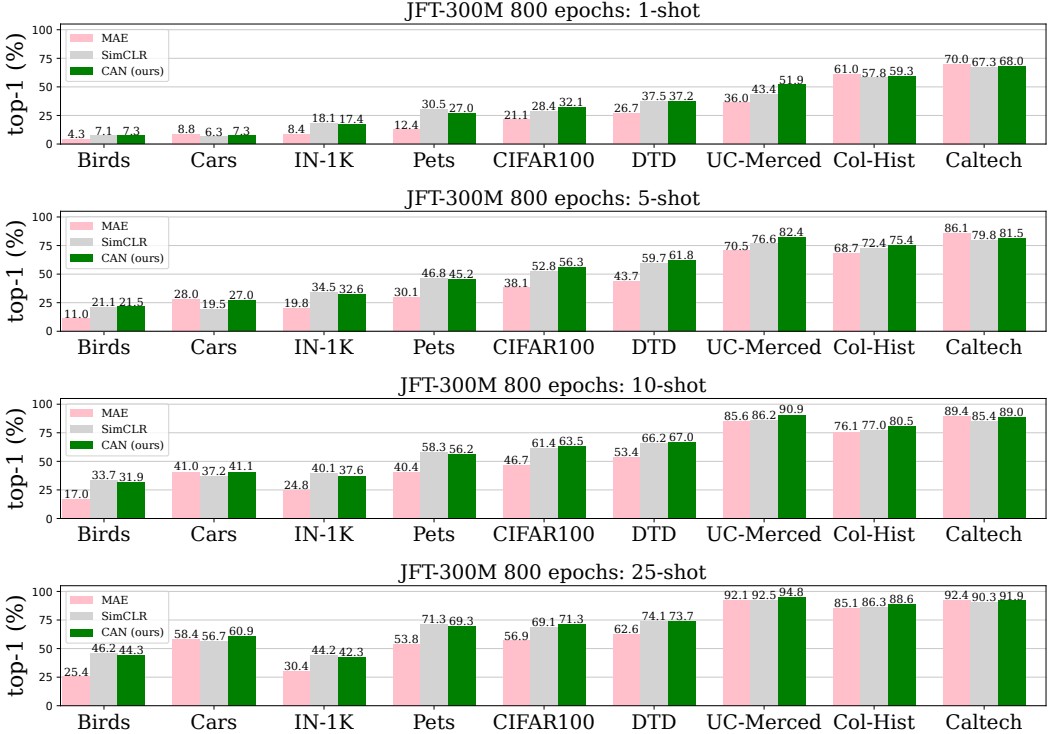

Figure 11: **Few shot:** ViT-L models pre-trained on JFT-300M for 800 epochs are evaluated on 9 few-shot learning tasks.

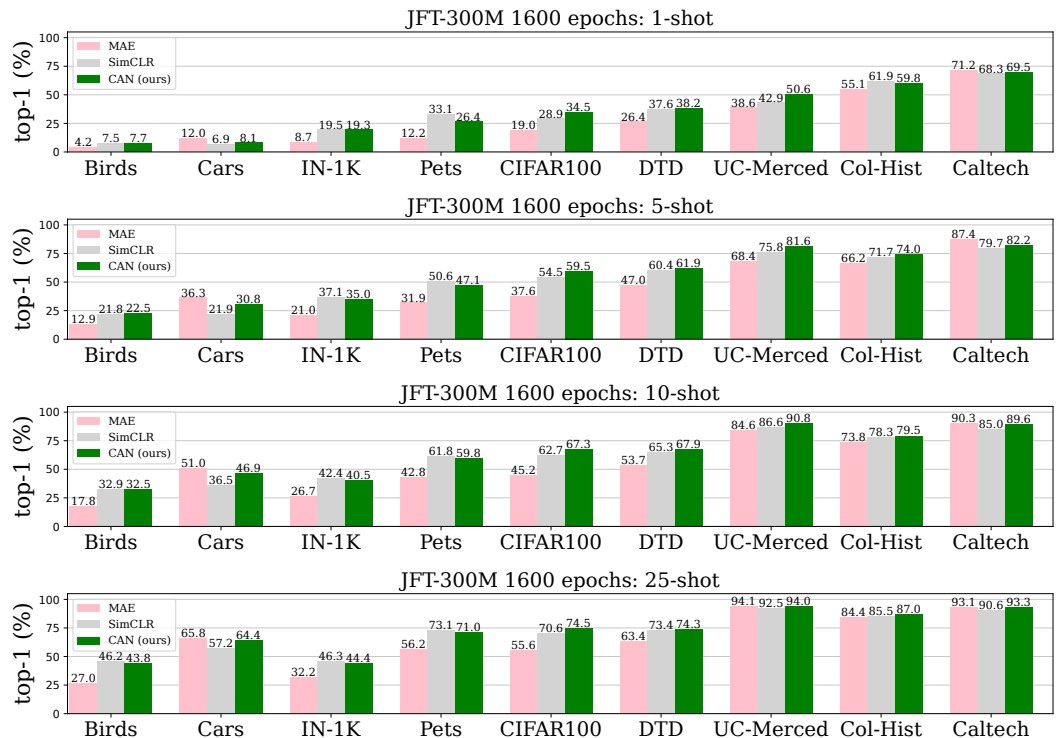

Figure 12: **Few shot:** ViT-L models pre-trained on JFT-300M for 1600 epochs are evaluated on 9 few-shot learning tasks.

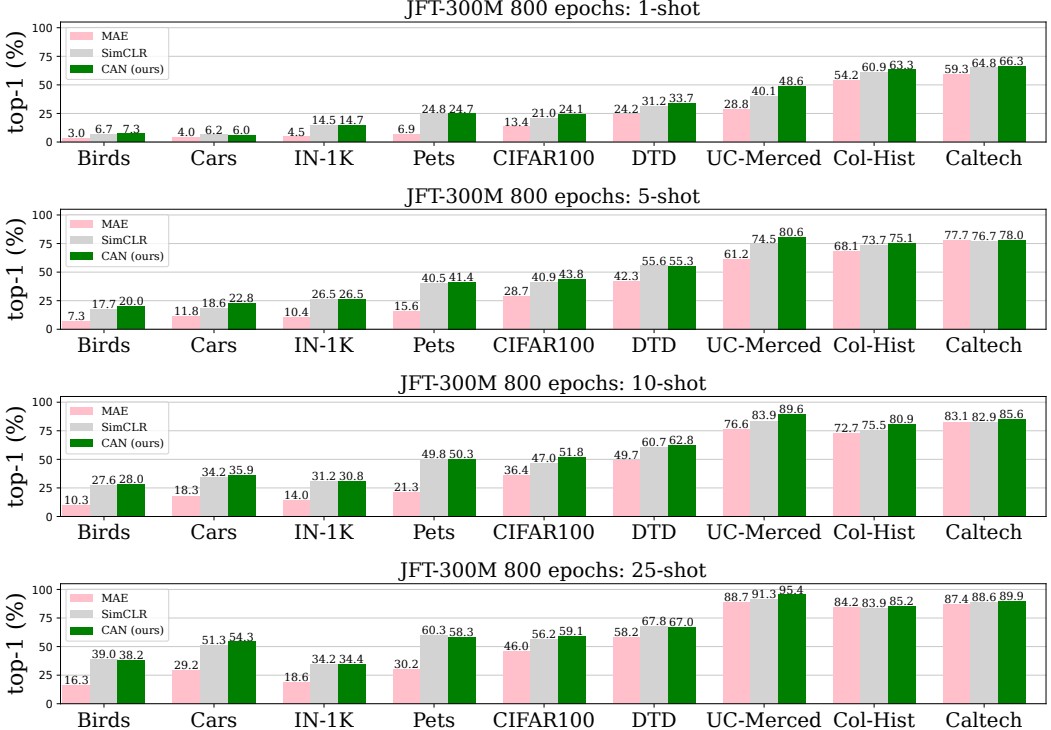

Figure 13: **Few shot:** ViT-B models pre-trained on JFT-300M for 800 epochs are evaluated on 9 few-shot learning tasks.

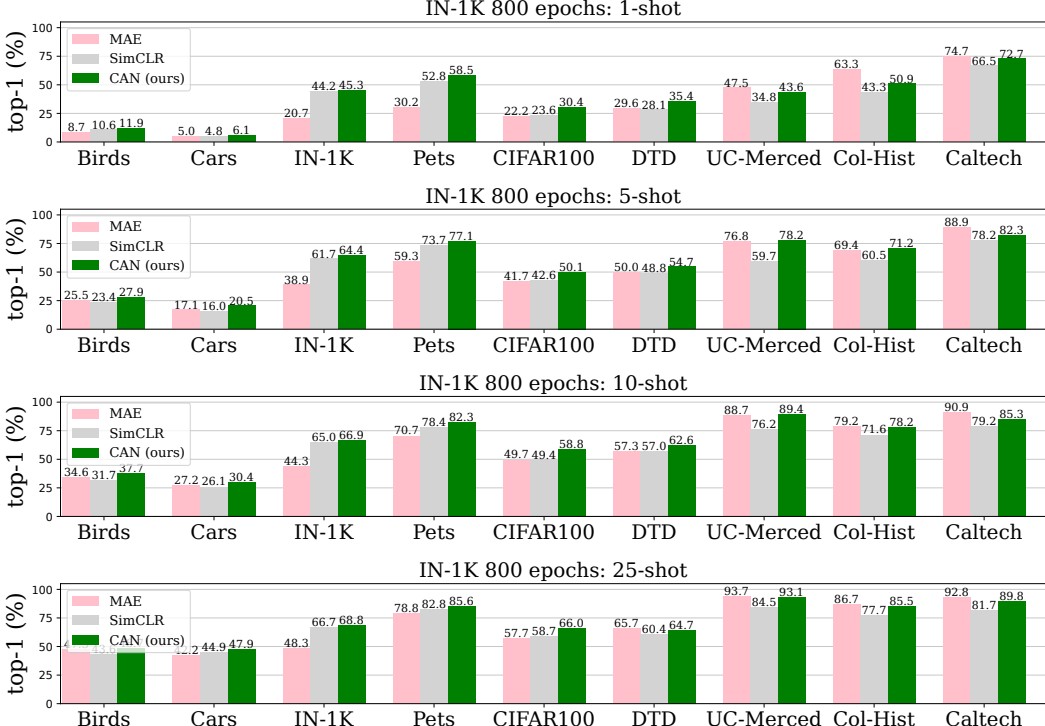

Figure 14: **Few shot:** ViT-L models pre-trained on IN-1K for 800 epochs are evaluated on 9 few-shot learning tasks.

We make a number of observations.

1. Across all settings CAN generally performs the best on JFT-300M pre-training.

2. The situation is less consistent on IN-1K pre-training. For instance, although MAE has comparatively poor few-shot performance on IN-1K, it is competitive on others for 25-shot evaluation: in this setting CAN only beats MAE on 5 out of 9 datasets. However, on 10-shot CAN outperforms MAE and SimCLR in 8 out of 9 cases, showing a subtle picture.

3. JFT-300M pre-training often outperforms IN-1K pre-training. Comparing Figures 11 and 14, JFT-300M yields better 25-shot CAN performance on 6 out of 9 datasets.

4. Model scale helps. Comparing Figures 13 and 11, ViT-L models perform best in nearly all cases.

## A.1  IMAGENET-21K PRE-TRAINING.

We also consider the performance of CAN on pre-training on ImageNet-21K (IN-21K), a publicly available dataset of 14.2 million images, grouped into 21,000 different classes Deng et al. (2009). We use the same hyperparamter settings as JFT-300M training to train ViT-L models on IN-21K for 800 (IN-1K equivalent) epochs.

We run a full set of evaluations on finetuning, linear probe, robustness (Figure 15), and few-shot learning (Figure 16.

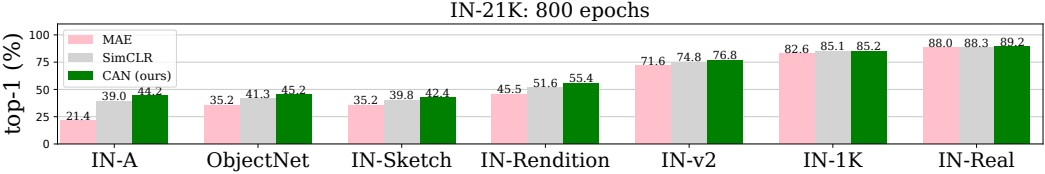

Figure 15: **Robustness:** ViT-L models pre-trained on IN-21K for 800 (IN-1K equivalent) epochs are first finetuned on IN-1K. The models are then evaluated on 7 test datasets with different distribution shifts from IN-1K.

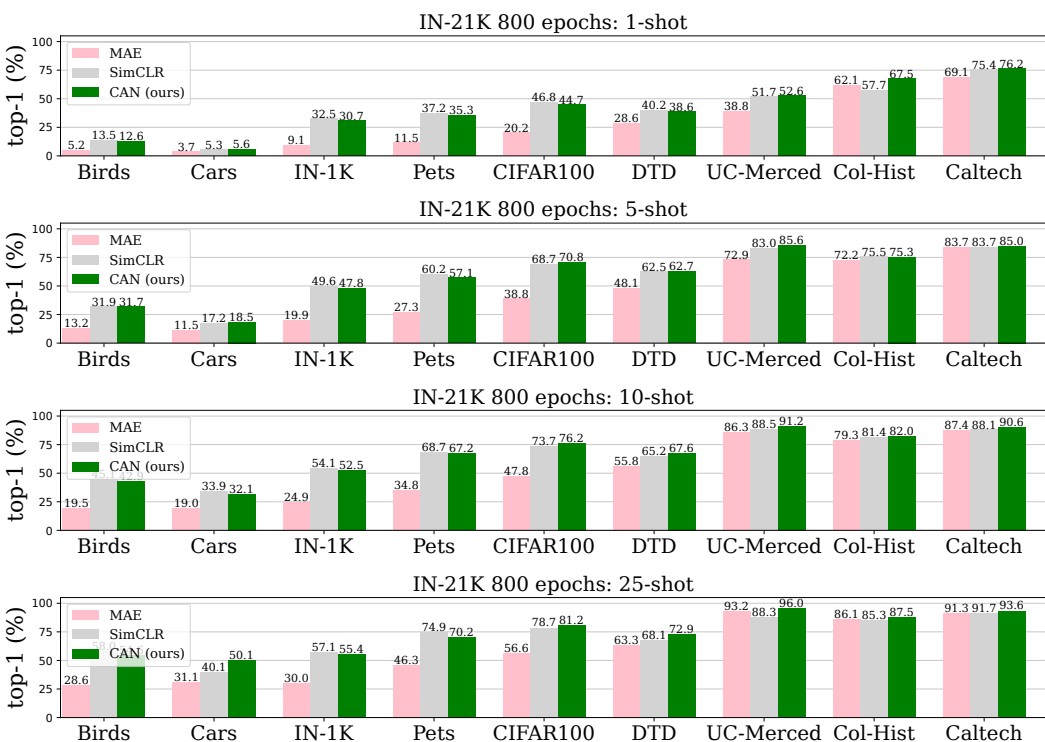

Figure 16: **Few shot:** ViT-L models pre-trained on IN-21K for 800 (IN-1K equivalent) epochs are evaluated on 9 few-shot learning tasks.

## B  RUNTIME OF CAN COMPARED TO DNC

In the main paper we estimate our method is significantly faster than DnC (Tian et al., 2021). We determined this approximate comparison from the following two pieces of information: 1) DnC reports that 3000 ImageNet epochs takes 29 hours on 512 TPUs for a ResNet-50 model ( 25M parameters), and 2) 3000 ImageNet epochs of CAN take 78 hours on 64 TPUs for a ViT-L model ( 300M parameters). We assume a linear relationship between number of TPUs and runtime. Under this assumption, we estimate that CAN would take approximately 10 hours to train with 512 TPUs, compared to the 29 hours reported by Tian et al. (2021) for a model with 1/10th the number of parameters. We emphasize that this is far from an exact comparison and is only intended as a very approximate guide.

## C  HYPERPARAMETER SETTINGS

We list hyperparameters used for CAN pre-training in Table 6 and Table 7. For preprocessing we closely follow SimCLR Chen et al. (2020b). We use the same hyperparameters for SimCLR pre-

training. For MAE pre-training, we use the same hyperparameters as listed in He et al. (2022), except for the use of Glorot uniform initialization instead of LeCun initialization as done in He et al. (2022). We found that this provided better performance for our JAX-based MAE implementation. Table 10 lists the hyperparameters for finetuning evaluations. We use the same set of hyperparameters for each finetuning each pre-training method, and for both ViT-B and ViT-L model sizes. For linear probing we list the hyperparameters in Table 11 for which we followed the settings in He et al. (2022). We use global average pool of the final representation instead of the cls token.

**MAE longer training:** MAE pre-training for longer training (5000 epochs) on JFT becomes unstable after about 500k steps (training loss oscillates); this results in poorer fine-tuning performance. To overcome this, we decrease the base learning rate by $75\%$ as shown in Table 9. However our model CAN is more stable and we use the same hyperparameters across different numbers of epochs.

**Few shot training:** For few-shot learning we use the same hyperparameters and pipeline as Dosovitskiy et al. (2021a). We use the same pre-processing as was done in (Kolesnikov et al., 2020). We use a base learning rate of 0.01 and train for 2500 steps, using an input resolution of $384 \times 384$.

**Hardware details:** We use TPU-v4 for all of our experiments. CAN on ViT-B uses 64 TPUs for a batch size of 4096. SimCLR, on the other hand, uses 128 TPUs for the same batch size, and is more compute intensive than CAN.

**Decoder architecture:** Our decoder architecture is the same as He et al. (2022). We use standard ViT with a decoder depth of 8 and decoder width of 512. We use 16 heads and 2048 as the dimension of the MLP.

**Projection head architecture:** We use 2 hidden layers in our projection heads. Each layer has a Fully-Connected (FC) layer (dim 4096) followed by BatchNorm (momentum=0.9) followed by ReLU. After these 2 layers we have a FC layer which transforms the features to 128 dimensions. We apply contrastive learning on top of these 128 dimensional features.

**JFT-300M specific hyperparameters:** All hyperparameters were determined by training on IN-1K, and directly transferred with JFT-300M pre-training, with the exception of learning rate and weight decay, which found needed to be at a lower level for JFT-300M. For all methods we divided the learning rate by a factor of $4$, and the weight decay by a factor of 2, except for MAE where we found that the original weight decay tuned on ImageNet worked better. Specifically, for CAN and SimCLR we used following parameter choices: $wd = 0.1/2 = 0.05$ and $lr = 1.25 \times 10^{-4}/4 = 3.125 \times 10^{-5}$ and for MAE we used $lr = 1.5 \times 10^{-4}/4 = 3.75 \times 10^{-5}$, and tried $wd = 0.05/2 = 0.025$, but found that the original $wd = 0.05$ worked better, so kept this value.

## C.1 CAN AND SIMCLR HYPERPARAMETERS

| config | value |
|---|---|
| optimizer | AdamW (Loshchilov & Hutter, 2017a) |
| base learning rate(ViT-B) | 2.5e-4 |
| base learning rate (ViT-L) | 1.25e-4 |
| weight decay (ViT-B) | 0.05 |
| weight decay (ViT-L) | 0.1 |
| optimizer momentum | $\beta_1, \beta_2 = 0.9, 0.95$ (Chen et al., 2020a) |
| batch size | 4096 |
| learning rate schedule | cosine decay (Loshchilov & Hutter, 2017b) |
| warmup epochs (Goyal et al., 2017) | 40 |
| augmentation | RandomResizedCrop, Color Jittering(strength=1.0), GrayScale(probability=0.2), Gaussian Blurring (probability=0.5) |

Table 6: Hyperparameters for CAN pre-training on ImageNet. Note that we use lower learning rate for ViT-L as compared to ViT-B, following Steiner et al. (2021). We use the same hyper-parameters for SimCLR training.

| config | value |
|---|---|
| optimizer | AdamW (Loshchilov & Hutter, 2017a) |
| base learning rate (ViT-B) | 2.5e-4 |
| base learning rate (ViT-L) | 3.125e-5 |
| weight decay | 0.05 |
| optimizer momentum | $\beta_1, \beta_2 = 0.9, 0.95$ (Chen et al., 2020a) |
| batch size | 4096 |
| learning rate schedule | cosine decay (Loshchilov & Hutter, 2017b) |
| warmup epochs (Goyal et al., 2017) | 40 |
| augmentation | RandomResizedCrop, Color Jittering(strength=1.0), GrayScale(probability=0.2), Gaussian Blurring (probability=0.5) |

Table 7: Hyperparameters for CAN pre-training on JFT-300M. Note that we use lower learning rate for ViT-L as compared to ViT-B, following Steiner et al. (2021). We use the same hyper-parameters for SimCLR pre-training.

## C.2 MAE HYPERPARAMETERS

| config | value |
|---|---|
| optimizer | AdamW (Loshchilov & Hutter, 2017a) |
| base learning rate (ViT-L) | 1.5e-4 |
| weight decay | 0.05 |
| optimizer momentum | $\beta_1, \beta_2 = 0.9, 0.95$ (Chen et al., 2020a) |
| batch size | 4096 |
| learning rate schedule | cosine decay (Loshchilov & Hutter, 2017b) |
| warmup epochs (Goyal et al., 2017) | 40 |
| augmentation | RandomResizedCrop, Color Jittering(strength=1.0), GrayScale(probability=0.2), Gaussian Blurring (probability=0.5) |

Table 8: Hyperparameters for MAE pre-training on IN-1K. We follow the choices made by He et al. (2022).

| config | value |
|---|---|
| optimizer | AdamW (Loshchilov & Hutter, 2017a) |
| base learning rate (ViT-L) | 3.75e-5 |
| weight decay | 0.05 |
| optimizer momentum | $\beta_1, \beta_2 = 0.9, 0.95$ (Chen et al., 2020a) |
| batch size | 4096 |
| learning rate schedule | cosine decay (Loshchilov & Hutter, 2017b) |
| warmup epochs (Goyal et al., 2017) | 40 |
| augmentation | RandomResizedCrop, Color Jittering(strength=1.0), GrayScale(probability=0.2), Gaussian Blurring (probability=0.5) |

Table 9: Hyperparameters for MAE pre-training on JFT-300M with ViT-L models. The only difference from the IN-1K configuration is the learning rate, which we reduced since we found tarining to be unstable.

## C.3 FUNETUNING AND LIENAR PROBE HYPERPARAMETERS

| config | value |
|---|---|
| optimizer | AdamW (Loshchilov & Hutter, 2017a) |
| base learning rate | 5e-4 |
| weight decay | 0.005 |
| optimizer momentum | $\beta_1, \beta_2 = 0.9, 0.999$ (Chen et al., 2020a) |
| batch size | 1024 |
| learning rate schedule | cosine decay (Loshchilov & Hutter, 2017b) |
| warmup epochs (Goyal et al., 2017) | 5 |
| training epochs | 100 |
| label smoothing | 0.1 |
| drop path | 0.1 |
| layer-wise lr decay | 0.65 |
| augmentation | RandomResizedCrop, Flip, RandAug(layers=2, magnitude=9) (Cubuk et al., 2020), Random Erase (Zhong et al., 2020)(probability=0.25) |

Table 10: Hyperparameters for finetuning CAN pre-trained model on ImageNet. We use the same hyperparameters for ViT-B and ViT-L, for both JFT-300M and ImageNet pre-trainined models.

| config | value |
|---|---|
| optimizer | LARS (You et al., 2017) |
| base learning rate | 0.1 |
| weight decay | 0 |
| optimizer momentum | 0.9 |
| batch size | 16384 |
| learning rate schedule | cosine decay (Loshchilov & Hutter, 2017b) |
| warmup epochs (Goyal et al., 2017) | 10 |
| training epochs | 100 |
| batch norm momentum | 0.9 |
| label smoothing | 0 |
| augmentation | RandomResizedCrop |

Table 11: Hyperparameters for linear probing for CAN pre-trained model on ImageNet. We use the same hyperparameters for ViT-B and ViT-L, for both JFT-300M and ImageNet pre-trainined models. Note that these hyperparamters are same as reported in He et al. (2022).

