# OpenReview forum: "CAN: A simple, efficient and scalable contrastive masked autoencoder framework for learning visual representations"
_ICLR.cc/2023/Conference — Submitted to ICLR 2023_

### Official Review · Reviewer_y8Yc · 2022-10-21

**Confidence:** 5
**Correctness:** 3
**Technical Novelty And Significance:** 3
**Empirical Novelty And Significance:** 2
**Recommendation:** 5

**Clarity, Quality, Novelty And Reproducibility:**

Well-written and part of the idea is interesting.
 However, the Title and some of the details (e.g., limitations, training costs, etc.) need to be reorganized and discussed.

**Strength And Weaknesses:**

Strength:
Interesting idea to add the diffusion noise prediction.
Good results for the linear probe.
Clear written.

Weaknesses:
(1) The Title of the paper does not cover the meaning of “noise” prediction accurately since it’s a synthesis of contrastive learning, masked autoencoders, and noise prediction.
(2) Choosing SimCLR as the primary baseline in Table 2 and the following Figures is actually not convincing. SimCLR is an early contrastive learning-based method and its performance has lagged behind other contrastive learning methods. How about DINO and MOCO v3?
(3) In Figure 6, masking 50% of patches is not the best choice, why the paper still uses 50%? The authors should better provide more explanations and quantitative analysis.
(4) How about the ablation study between contrastive learning loss, reconstruction loss, and denoise loss? (That’s the result of a pairwise combination)
(5) Lack of downstream experiments on Object detection and Semantic segmentation.
Typos:
Page 4: “see Table 1” --> “see Figure 1”


**Summary Of The Paper:**

The paper proposes a visual representation learning framework combining contrastive learning, masked autoencoders, and diffusion noise prediction. It is interesting to introduce noise prediction for the unmasked tokens. The models pre-trained by the framework achieve competitive results on the image classification task.

**Summary Of The Review:**

I am inclined to accept the paper if the author can dispel my concerns well.

---

> ### Author Response · Authors · 2022-11-14
> **Response to reviewer y8Yc**
>
> Thank you for the time taken to review our work, we sincerely appreciate it, and are glad you found the paper to be presenting an interesting idea with good downstream performance.  We are also glad you found our work to be clearly written.
>
> You mentioned your inclination to accept the paper if we address your concerns, which we would like to take the time now to do.
>
> ---
>
> > **Ablation study for C, A, and N**
>
> We have the comparisons of the C (contrastive), A (masked autoencoder) and N (de-Noising) losses you ask for. Our setting in ViT-B pre-trained for 100 epochs on ImageNet, evaluated with a linear probe on ImageNet:
> - C and A and  N (full method): 68.9
> - C and A: 67.9
> - A and N: 42.8
> - C and N: 68.5
>
> We will prepare longer runs for the final paper.
>
> ---
>
> > **Why 50% masking?**
>
> We agree that a lower masking rate could be used to get better performance, but this would be a more expensive method. **A key focus of this work is in designing a method that is both efficient and highly performant (Figure 1)**. We decided to use 50% masking since we felt the efficiency improvement outweighed the performance difference.
>
> ---
>
> > **Why SimCLR as primary baseline?**
>
> Note that the contrastive part of CAN uses SimCLR too. Any other contrastive method could be used instead as part of CAN. We compare CAN to a SimCLR baseline since we wish to show that CAN outperforms its constituent contrastive and masked autoencoder parts.  Further, while other methods have advanced the state-of-the-art on ImageNet, SimCLR is still a competitive algorithm for pre-training on Web-scale data like JFT.  E.g., this paper [https://arxiv.org/pdf/2209.15589.pdf] ] finds that:
>
> - SimCLR is more scalable than BYOL and DINO (Sec 3.3).
> - SimCLR enjoys a better efficiency-performance tradeoff compared to BYOL, DINO (Fig 1 and 2).
>
> Since SimCLR is widely used, simple, and competitive at large scales, we made the choice to use SimCLR throughout.
>
> ---
>
> > **Object detection and segmentation results?**
>
> We are working on object detection results for the final paper. But we would also like to highlight the extensiveness of our downstream evaluations. We evaluate few-shot and robustness to distribution shift tasks in many different settings (pre-training on different datasets, different models, different epochs).
>
> ---
>
> > **Comparison of training costs?**
>
> Fig 1 (left) compares the FLOPs in CAN, SimCLR, and MAE forward passes. We found this corresponds closely to wallclock time, with the 5000 epoch ViT-L training taking around 2 weeks with SimCLR, 6 days with CAN, and 4 days with MAE.

---

> > ### Author Response · Authors · 2022-11-17
> > **Happy to give further clarifications**
> >
> > Thank you for your time and efforts in reviewing our paper! We sincerely hope our response successfully addresses your concerns and answers your questions. Since there are two days left of the rebuttal period, we are very eager to know if there are any further concerns or questions we can address, so please do let us know if there are.
> > Thank you again for the constructive suggestions you have already given. They have helped improve our paper.
> >
> > Best,\
> > The Authors

---

### Official Review · Reviewer_JaaX · 2022-10-23

**Confidence:** 5
**Correctness:** 3
**Technical Novelty And Significance:** 3
**Empirical Novelty And Significance:** Not applicable
**Recommendation:** 5

**Clarity, Quality, Novelty And Reproducibility:**

The idea is novel with extensive empirical studies. There is no code for reproduction.

**Strength And Weaknesses:**

Strength:
+ Combining contrastive learning, masked autoencoders, and noise prediction is novel.
+ Extensive empirical studies on linear evaluation, fine-tuning, transfer learning, and robustness demonstrate the effectiveness of the proposed method.

Weaknesses:
+ JFT-300M is not accessible to everyone. Thus, ImageNet-1K or ImageNet-22K is more suitable for researchers to conduct fair comparisons. It seems the performance of CAN is not superior to concurrent work CAE or even MAE in terms of fine-tuning accuracy.
+ Lack of full ablation studies for each component.
+ For the denoising part, how is MLP designed? Why is the target noise? How could you ensure the network could learn high-frequency information as claimed?

**Summary Of The Paper:**

This paper proposes CAN, a simple, efficient, and scalable method for self-supervised learning of visual representations, which combines contrastive learning, masked autoencoders, and noise prediction approach used in diffusion models. The learning mechanisms are complementary to one another. Extensive empirical studies on linear evaluation, finetuning, transfer learning, and robustness demonstrate that the proposed method achieves strong downstream performance.

**Summary Of The Review:**

See Strength And Weaknesses

---

> ### Author Response · Authors · 2022-11-14
> **Response to reviewer JaaX: new ImageNet-21k results and ablations**
>
> Thank you for your time taken to review our work, we truly appreciate your efforts, and are really happy to see you liked the novelty of our pre-training task, and the efforts we made to evaluate CAN on as many different downstream tasks as possible.
>
> We believe that these positive points can form the basis for an eventually positive overall assessment of our work. Your review noted a couple of concerns, which we believe that we can address in a concrete way with new experimental results.
>
> ---
>
> > **Results on public datasets**
>
> As you suggest we have run experiments on ImageNet-21k, a large-scale publicly available dataset. Our setting is ViT-L models pre-trained for 800 epochs (ImageNet-1k equivalent epochs). We run a full set of evaluations on ImageNet-1k linear probe, finetune, robustness, and few-shot learning. See Appendix A.1 for all the new results, but for instance these are our robustness (and IN-1K linear probe) results:
>
>
> | Dataset| IN-1K (finetune) | IN-1k (lin. probe) | IN-Real | IN-A | IN-R | Sketch | IN-v2 | ObjectNet |
> | ----------- | ----------- | ----------- |----------- |----------- |----------- |----------- |----------- |----------- |
> | MAE | 82.6  | 57.8 | 88.0 | 21.4 | 45.5 | 35.2 | 71.6 | 35.2 |
> | SimCLR | 85.1 | 76.5 | 88.3 | 39.0 | 51.6 | 39.8 | 74.8 | 41.3 |
> | CAN (ours) | 85.2 | 76.5 | 89.2 | 44.2 | 55.4 | 42.4 | 76.8 | 45.2 |
>
> We emphasize that CAN is about 50% more efficient than SimCLR in the ViT-L setting.
>
> Finally, we would also like to emphasize the strength of our existing public dataset ImageNet-1k results. We outperform MAE and SimCLR on many tasks including few-shot prediction, and robustness to distribution shift. We think these results are very important: few-shot in particular is arguably the setting in which self-supervision is most needed. We do not think that the performance of a method should be reduced to a single number (the ImageNet-1k finetune number). Indeed the fact that CAN and MAE have the same finetuning but very different performances on other downstream tasks highlights exactly why this is important.
>
> ---
>
> > **Ablation of C, A and N**
>
> We have the comparisons of the C (contrastive), A (masked autoencoder) and N (de-Noising) losses you ask for.  We evaluate ViT-B models pre-trained for 100 epochs on ImageNet with a linear probe on ImageNet:
> - C and A and  N (full method): 68.9
> - C and A: 67.9
> - A and N: 42.8
> - C and N:  68.5
>
> We note that only the final number is new: the first two numbers are from Table 1, the third from Figure 7 (left). However, we agree that the ablation results are not organized clearly, so we have updated the paper to give these results all in one table (Table 5).
>
> ---
>
> > **The denoising method**
>
> **What is the MLP?** It is a simple two layer ReLU MLP  with input, width, and output dimension all equal to the embedding dimension of the ViT encoder (768 in the case of ViT-B). We have added this information to the paper. We tried different depths and widths, and found performance to be robust to these choices.
>
> **Why predict noise?** We are motivated by the denoising loss used to train diffusion model: in this literature predicting the noise is standard practice, so we adopt the same (See Sec 3.4 for further discussion). We tested using the clean input as target instead of noise, but found that it didn’t work well (corroborating similar findings in the diffusion literature).

---

> > ### Author Response · Authors · 2022-11-17
> > **Happy to give further clarifications**
> >
> > Thank you for your time and efforts in reviewing our paper! We sincerely hope our response successfully addresses your concerns and answers your questions. Since there are two days left of the rebuttal period, we are very eager to know if there are any further concerns or questions we can address, so please do let us know if there are.
> > Thank you again for the constructive suggestions you have already given. They have helped improve our paper.
> >
> > Best,\
> > The Authors

---

### Official Review · Reviewer_sdad · 2022-11-04

**Confidence:** 3
**Correctness:** 4
**Technical Novelty And Significance:** 3
**Empirical Novelty And Significance:** 3
**Recommendation:** 8

**Clarity, Quality, Novelty And Reproducibility:**

**Quality**: The paper has concrete motivation. I would suggest changing the introduction though a bit and starting directly from the benefits of combining the self-supervised approaches rather than from the properties of the datasets (curated vs uncurated).  I think that could help getting to the key idea and setting the relevant context from the beginning.

**Clarity**: The paper is very well-written, the idea is clearly presented, especially the modeling section in terms of technical descriptions, and it is accompanied with useful diagrams and visualizations. The experiments section is straightforward and easy to follow. I think in the presentation it will be good to make some of the expressions in paragraphs such as the first and second, and the first of the 5th page, centered in their own line rather than packed with the text, to make it easier to follow the modeling section.

**Novelty**: Combining the approaches of contrastive learning and masked auto-encoding is simple but novel, and seems to reflect parallel discoveries in the NLP domain (with BERT). The diffusion-inspired noise-prediction loss is also novel. The related work section of the paper is also good and covers the pillars this works builds upon: masked autoencoding, contrastive learning and denoising diffusion models.

**Reproducibility**: The approach is described to a sufficient degree, and the hyperparameters impact is analyzed in the paper, covering  masking rates, noise level and loss weights, and the selected hyperparameters are detailed in the appendix.


**Strength And Weaknesses:**

**Strengths**:
* **Combining complementary approaches**: the paper combines contrastive learning with masked auto-encoding to achieve complementary benefits of global and local signals. In high-level it’s similar to BERT two losses at the sentence level and at the masked-word level. That’s a great simple idea that that paper shows to work well. The usage of both noise prediction and auto-encoding also intuitively seem to play well together, where the former captures high-frequency features while the latter captures lower ones.

* **Model Technical details**: I also like the specific technical manner in which the ideas are combined and find it an conceptually appealing approach (like a form of function composition): two masks are produced and then the two partial views are compared using contrastive learning. If the representations are good, then two partial views of the same image should be close to each other compared to other images.

* **Novelty**: The ideas explored in the paper are novel and simple. It combines existing general methods (contrastive learning, masked autoencoding) in an original manner making them complement each other. See more details below.

* **Simplicity**: The way the two approaches are combined is simple and elegant, and so given the good empirical results, more likely to be broadly adopted by the community. In such cases, simple ideas are preferable in my opinion than more complex combinations as is the case with other works that explore manners to combine these approaches.

* **Extensive set of suitable experiments**: multiple experiments are presented over both ImageNet and larger-scale JFT datasets, and ablations are also explored. The contribution of each loss is being explored and it looks like each of them contributes to a comparable degree. The subject of efficiency is also explored in its own section, although extending it further with additional experiments could be useful. Finally, the paper also explores the model’s robustness under distribution shifts. Additional experiments in the appendix explores additional transfer learning experiments.

* **Strong results and suitable baselines**:  strong results on the imagenet linear probing compared to leading approaches such as MAE and SimCLR. It also had a lot better computational efficiency than SimCLR.

**Weaknesses**:

I don’t identify fundamental weaknesses. See some suggestions for writing and presentation improvement below. Smaller improvements:
* **Limitations and future directions discussion**: Would be good if the authors could discuss any limitations they recognize of the work and potential directions for next steps!
* **Qualitative results and visualizations**: could be good to show any sort of qualitative results and some image samples, or empirical visualizations of the model.
* **More baselines**: Comparing to additional baselines will also be useful to corroborate the paper findings.


**Summary Of The Paper:**

The paper presents a new approach called CAN that combines contrastive learning and masked autoencoders, and also adds denoising diffusion-inspired loss to learn visual representations. Experiments are performed over Imagenet where the model is compared and surpassing leading self-supervised approaches like MAE and SimCLR. The approach is also much more computationally efficient than SimCLR.

**Summary Of The Review:**

That’s a great paper with a novel and simple idea that combines proven modern approaches and shows strong results across multiple empirical dimensions (accuracy, efficiency, few-shot learning, robustness under distribution shifts) compared to leading methods on an important task of self-supervised visual learning. I recommend acceptance.

---

> ### Author Response · Authors · 2022-11-14
> **Response to reviewer sdad**
>
> We are grateful for volunteering your time to review our work. We are sincerely thankful you appreciate the significance of our work, including the novelty of the method, its simplicity, and its empirical performance. We humbly ask you to ensure these points are recognized during the reviewer discussions.
>
> We would like to respond to comments and questions asked in your review.
>
> ---
>
> > **Discussing limitations**
>
>  We agree that this is both an important and good idea. Some limitations we will include are:
> - The loss weights between contrastive and reconstruction needs tuning. This may be problem dependent, however for ImageNet, ImageNet-21k and JFT we found the same setting worked well.
> - More future work than a limitation: it would be interesting to systematically study how to combine different pre-training tasks, and which are complementary and which aren’t.
>
> ---
> > **Clarity**
>
> Thank you for the constructive suggestions on how to improve the readability of our work. We will incorporate your suggestions.
>
> ---
>
> > **Additional results**
>
> Finally, although this wasn't asked for in your review, we would like to bring your attention to additional experiments we have produced in response to the comments of other reviewers. The main new results are for pre-training on ImageNet-21K. We trained ViT-L models of 800 epochs and ran a full set of evaluations: linear probe, finetuning, few-shot, and robustness. We have included these new results in Appendix A.1.

---

### Official Review · Reviewer_Jkgq · 2022-11-04

**Confidence:** 3
**Correctness:** 3
**Technical Novelty And Significance:** 3
**Empirical Novelty And Significance:** 3
**Recommendation:** 5

**Clarity, Quality, Novelty And Reproducibility:**

The writing was clear, but some of the results do not seem reproducible (e.g. the MAE results). Also, I had a few questions:

- What is the virtue of symmetry between the two backbones? It seems that breaking symmetry between the encoders can have benefits.
- Why 50% masking ratio? MAE showed that higher ratios achieved better performance.
- Are you comparing against an MAE with a 50% masking ratio in table 2?
- Why is there no FLOP comparison with MAE when the abstract reports the FLOP improvement over SimCLR? I suspect this method is not more FLOP efficient, but the authors need to at least report that result and explain why the results are that way.

**Strength And Weaknesses:**

Strengths
- Combining the denoising diffusion objective with masked reconstruction in order to improve the prediction of high-frequency features without incurring significant additional computation cost is a good idea. This reviewer would be curious to see how much this approach benefits other SSL methods independently of the CAN method.
- The experimental infrastructure is impressive -- the experiments required a lot of hardware time to run.

Weaknesses
- In my opinion, the primary weakness of this paper lies with its comparison to baselines. It’s unclear why the authors' implementation of MAE appears to differ so significantly from the original method (e.g., different masking ratio,  different image augmentations, different learning rate). For example, the MAE paper explicitly mentions that their method does not use color jitter since it degrades performance, yet in this paper the authors apply color jitter during pretraining.
- It seems odd to report only the linear probe accuracy in Figure 1, without the finetuning performance, when including that information seems like it would paint a different (and more complex) picture. Further, as mentioned in [1], linear probing performance does not necessarily correlate well with transfer learning performance (i.e., finetuning well on a downstream task of interest is usually more valuable than linear separability of the learned features). And in addition, Figure 5 in the MAE paper [2] shows a significant difference in the linear probe performance vs the finetuning performance when masking rate is changed. They observed that linear probe accuracy varied quite significantly, while the finetuning performance was more robust to choice of hyperparameters. This reviewer is open to the possibility that their understanding is incorrect, but in that case phrases in the reviewed paper such as “In experiments our default masking rate is m = 50% unless explicitly stated otherwise.” are misleading.
- The authors do not provide the source code (though they promise they will release it). This would have been useful to clear up any misunderstandings about their implementation of the MAE.
- On page 8, the authors claim that “CAN performs particularly well under JFT-300M pre-training: it improves over SimCLR and MAE baselines in all 7 cases, often by significant margins”. However, their results in Figure 5 do not support this claim. MAE appears to have outperformed CAN in 3 of the 7 cases when pre-trained on JFT-300M and finetuned on IN-Real, IN-Sketch, and IN-V2.
- Table 2 and 3 seem to be a bit misleading, or at least the work is lacking some citations. While the comparison to exemplar methods of these SSL families is important, acknowledgement of the other methods is also important, especially if comparing to the “state of the art” on ImageNet linear probe. For example, [3] is an ICLR 2022 paper  that shows several methods with comparable FLOP and parameter counts to CAN, but that achieves better linear probe performance. More specifically, EsVIT with the Swin-B architecture achieved over 80% on the ImageNet linear probe task, and the EsViT with Swin-T has only 28 M parameters and achieved 78.1% linear top-1 acc on ImageNet, whereas the best reported method in Table 3 achieved 78.2%, and CAN achieved 76.2%.

[1] Exploring Simple Siamese Representation Learning https://arxiv.org/pdf/2011.10566.pdf
[2] Masked Autoencoders Are Scalable Vision Learners https://arxiv.org/pdf/2111.06377v2.pdf
[3] EFFICIENT SELF-SUPERVISED VISION TRANSFORMERS FOR REPRESENTATION LEARNING https://arxiv.org/pdf/2106.09785.pdf


**Summary Of The Paper:**

CAN is a method of self-supervised learning for visual representations. The method builds on three families of self-supervised learning methods: Contrastive learning, Masked Autoencoding, and Noise prediction. The paper claims that the learning mechanisms of each family are complementary to one another, and includes some experiments with comparisons to selected methods in the masked autoencoding family (MAE) and the contrastive learning family (SimCLR). The paper claims empirical improvements over MAE and SimCLR on ImageNet, and when pre-training on JFT-300M and fine-tuning on ImageNet they achieve an 85.9% top-1 accuracy.


**Summary Of The Review:**

The method seems promising and the topic is important and timely. However, the paper failed to make convincing comparisons (e.g., to EsViT, or to a strong MAE baseline with a fair masking ratio and FLOP comparison). In addition, the paper did not present experiments that illuminated the effects of combining masking, denoising, and contrastive learning methods in general, but only in the context of their method, CAN, and when comparing to their own MAE implementation. Further, the paper lacked several important details, and as a result this reviewer lost confidence in the efficacy of the proposed method.

---

> ### Author Response · Authors · 2022-11-14
> **Response to reviewer  Jkgq: clarifying critically important details (1/2)**
>
> We are grateful for the time taken to review our work, and for appreciating contributions such as the value of the noise prediction task. In the summary of your review you mention two main reasons for your negative assessment: 1) uncertainty over the MAE baseline implementation, 2) Omitting a study of the effects of the three loss terms.
>
> We humbly ask you to carefully consider the clarifications below, since we believe they should lead you to important reassessments of our work.
>
> ---
>
> **MAE baseline:**
> We were very careful in our tuning of MAE, and there has been some unfortunate confusion which we explain now:
> - **Masking rate: we use 75% as in the MAE paper.** The sentence “our default masking rate is m = 50% unless explicitly stated otherwise” refers only to CAN, not to MAE. We have updated the paper to clarify this sentence. In fact we tuned the masking rate ourselves on ImageNet and JFT, finding in both cases that 75% was optimal, reproducing the finding of the original MAE paper.
>  - **Augmentation: we use the same augmentations as in the MAE paper.** An unfortunate typo in the appendix (Table 7) says that we use jitter etc. too, which we believe to be the reason for the confusion. We apologize for this human error. We have added our MAE configs to the supplementary for you to view.
> - **Learning rate: we use the same IN-1K learning rate as the MAE paper.** The different learning rate in the appendix is for JFT pre-training: in this setting we cut the IN-1K learning rates for MAE, SimCLR and CAN by a factor of 4 since training was unstable otherwise.
> - In general, we carefully match all the parameters in the official MAE code. Results in Table 3 show that our implementation approximately matches the official MAE reported results (modulo PyTorch vs JAX differences).
>
> ---
>
> **Ablating the three components of CAN:**
>
> We have the comparisons of the C (contrastive), A (masked autoencoder) and N (de-Noising) losses you ask for. Our setting for ablations is ViT-B pre-trained for 100 epochs on ImageNet evaluated with a linear probe on ImageNet-1k:
>  - C and A and  N (full method): 68.9
> - C and A: 67.9
> - A and N: 42.8
> - C and N:  68.5
>
> We note that only the final C & N number is new: the first two numbers are from Table 1, the third from Figure 7 (left). However, we agree that the ablation results were not organized clearly, so we have updated the paper to give these results all in one table (Table 5). We will produce more ablations (more epochs, JFT pre-training etc.) for the final paper.

---

> > ### Author Response · Authors · 2022-11-14
> > **Response to reviewer Jkgq (2/2)**
> >
> > This message addresses other points raised in your review. Please note we made actionable changes to the paper where possible:
> >
> > > **Comparisons in Table 2 and 3:**
> >
> >  We have added EsViT to Table 3 in the manuscript. However we would like to emphasize that Swin-T is not a comparable architecture to ViT: e.g. the official Swin-T supervised ImageNet performance is 81.3% compared to official ViT-B performance of 77.9%, while the Swin-T image throughput is 755 im/s compared to ViT-B 86 im/s. So fair performance and efficiency comparisons between CAN and EsViT are very difficult. We had made the original decision to only compare to ViT models for this reason.
> >
> > ---
> >
> > > **Why is there no FLOP comparison with MAE?**
> >
> > Fig 1 shows FLOPs for MAE.
> >
> > ---
> >
> > > **Only linear probe in Figure 1:**
> >
> > We have moved the finetuning plot to Figure 1 to give the full picture up front. CAN still gets best performance on both evaluations, and scales better with epochs for finetuning.
> >
> > ---
> >
> > > **Source code:**
> >
> >  We will release it with the final paper.
> >
> > ---
> >
> > > **“CAN improves in all 7 cases”:**
> >
> > Apologies, this sentence was false. We have since updated this  figure, and CAN is best in 6 out of 7 cases. The change we made was to the finetuning evaluation: we added early stopping to fix overfitting during finetuning.
> >
> > ---
> >
> > > **Why 50% masking?**
> >
> > We justify this choice in the final "Masking Rate" paragraph of Sec 5:, there is an efficiency-performance trade-off, with 50% masking being much more efficient for relatively little change in linear probe.

---

> > > ### Comment · Reviewer_Jkgq · 2022-11-15
> > > **Response to authors**
> > >
> > > Thank you for your response which, in my view, rectified some important typos and confusions in the draft.
> > >
> > > I appreciate the additional results in table 5 which seem to indicate that the contrastive learning signal is the dominant one in this formulation. This makes it much clearer to me how this method is working.
> > >
> > > In Table 3, is there a reason why you don’t pre-train for 1600 epochs with ViT-L? This seems like an important experiment to run given the results in the MAE paper in Table 3 that compare with previous results on ImageNet-1K are performed with 1600 epochs of pre-training.
> > >
> > > I see the new results in Figures 4 and 5. Could you provide an additional description of the early stopping procedure? The caption mentions that the reported values are validation performances, in which case this is not really early stopping in the way that most people in the community understand it, since that would imply that you are reporting the test accuracy for the model with the best validation accuracy.  Did you re-run the evaluation for all the models? I’m trying to understand why the numbers for SimCLR  and CAN changed in the top row (JFT-300M pre-train) of Figure 5 while the numbers for MAE remained constant between the drafts. Did the MAE just not overfit (and thus was not helped by the early-stopping protocol)? Could you provide additional analysis of the results in Figures 4 and 5? The results seem to have changed pretty dramatically on some datasets between drafts and far less on other datasets. What is your interpretation of the relative performances between the methods? What is your interpretation of the change in the performance of these methods between drafts?
> > >
> > > In light of your changes to the draft adding the finetuning results in Figure 1 and the resolution of some of our misunderstandings about the MAE, I have increased my estimate of the correctness and novelty from a 2 to a 3, and my overall recommendation from a 3 to a 5.

---

> > > > ### Author Response · Authors · 2022-11-15
> > > > **Clarifications on finetuning protocol**
> > > >
> > > >
> > > > Thank you very much for your response. We are sincerely grateful for the time taken to consider the clarifications to the original submission made in our response. We are also grateful for your willingness to reassess our work based on these discussions.
> > > >
> > > > Your response raises some further points, including clarifications on the changes to the finetuning protocol, which we would also like to address.
> > > >
> > > > ---
> > > >
> > > > > **Early stopping procedure**
> > > >
> > > > You had a number of questions about how we approached this, so let us explain it all in full detail.
> > > >
> > > > - First, we used the exact same pre-trained models. All that was changed was the evaluation. The early stopping only applied to full finetuning, so it is the robustness figures (Fig 5, 8, 9) that are affected.
> > > > - For all methods we adopted the exact finetuning hyperparameters as described in the MAE paper. For instance, using the same augmentations, epochs, weight decay, learning rate schedule and so on (as described in Table 9 of the MAE paper, and their official code). This includes details such as  applying BatchNorm before the final linear layer.
> > > > - We found this setup worked well for MAE-trained models, with the final finetuned model achieving the best validation accuracy.
> > > > - However, it led to overfitting during finetuning for CAN and SimCLR. We suspect that this is in part due to contrastive methods structuring their embedding space differently [as discussed in https://arxiv.org/pdf/2205.14141.pdf for instance]. It is therefore possible that a different finetuning setup could improve the finetuning for CAN and SimCLR further, although we did not pursue this.
> > > > - We noticed a consistent pattern for CAN and SimCLR: their IN-1K validation performance was peaking at 70k steps and 65k gradient steps respectively during finetuning.
> > > > - We therefore implemented a very simple modified finetuning protocol for each: stopping training after 70k and 65k steps respectively. This is what we meant by early stopping.
> > > > - To give a succinct overview: the MAE numbers stayed the same since the original finetuning protocol was tuned for MAE, and produced non-overfit MAE results. The CAN and SimCLR numbers are updated because we reduced the number of training steps for both, keeping all other aspects of the finetuning evaluation fixed.
> > > >
> > > > We hope this helps clarify the changes. We will update the paper to describe all of these details for other readers.
> > > >
> > > > ---
> > > >
> > > > > **Interpreting the new robustness results**
> > > >
> > > > You also asked about how to interpret the different changes in performance for different datasets. We note that the biggest changes (of order ~5%) are to the datasets with worst test performance {IN-A, ObjectNet, IN-Sketch} which is probably to be expected.
> > > >
> > > > Perhaps the most notable other change was to ImageNet-Redition [https://github.com/hendrycks/imagenet-r], where CAN improved from 61.1 to 67.1, and SimCLR from 56.4 to 61.9. It is hard to formally interpret this, but we note that this dataset is comprised of sketches, artwork, graffiti and so on. Our shorter schedule (“early stopping” procedure) is stopping the model from overfitting _to spurious features in the usual IN-1K dataset_. We speculate that these features may be high frequency details that are found in real-world photos. These features are less present in ImageNet-Redition images, which are often more clean cut. Our suggestion, which is only a conjecture, is that this could explain the difference in ImageNet-Redition performance.
> > > >
> > > > ---
> > > >
> > > > > **ImageNet-1k ViT-L 1600 epochs**
> > > >
> > > > Thank you for the actionable suggestion. We fully agree that this is a valuable experiment.We will prepare these numbers for the camera-ready version.

---

### Decision · Program_Chairs · 2023-01-20

**Decision:**

Reject

**Justification For Why Not Higher Score:**

Overall it is a reasonably good paper but seems not interesting enough to go beyond the high bar of a top ML conference.

**Justification For Why Not Lower Score:**

N/A

**Metareview: Summary, Strengths And Weaknesses:**

The paper proposes a self-supervised learning method, called CAN, for visual representation. The proposed idea is simply: to combine contrastive learning, masked autoencoding, and noise prediction. In experiments, the method outperforms MAE and SimCLR on ImageNet.

Strength. The combination of contrastive learning, masked autoencoding, and noise prediction is interesting. The empirical evaluation is extensive. The paper is also clearly written.

Weakness. Combining existing ideas and seeing the somewhat expected benefit is not much exciting. Lack of reproducibility as JFT-300M is not publicly available. Also, multiple reviewers raised concerns about the baselines (e.g., MAE performance).

Overall it is a reasonably good paper but seems not interesting enough to go beyond the high bar of a top ML conference.